# Comparison of the Declared and Simulated Real-Use Noise Data during Wood Sanding Using a Hand-Held Power Sander

Miroslav Dado [1], Marián Schwarz [2], Jozef Salva [2,*], Richard Jankovič [1] and Richard Hnilica [1]

1 Department of Manufacturing Technology and Quality Management, Faculty of Technology, Technical University in Zvolen, Študentská 26, 960 01 Zvolen, Slovakia; dado@tuzvo.sk (M.D.); xjankovic@is.tuzvo.sk (R.J.); hnilica@tuzvo.sk (R.H.)

2 Department of Environmental Engineering, Faculty of Ecology and Environmental Sciences, Technical University in Zvolen, T. G. Masaryka 24, 960 01 Zvolen, Slovakia; schwarz@tuzvo.sk

\* Correspondence: xsalvaj@tuzvo.sk

**Abstract:** The hand-held power sander is a frequently used tool in woodworking, and it is a significant source of risk in terms of dust, vibration, and, notably, noise. The purpose of a hand-held power sander manufacturer's noise emission statement is to provide information that is useful for assessing the risks associated with noise exposure and should assist users in selecting a hand-held power sander with reduced noise emissions. The stated levels of emitted noise obtained in accordance with a harmonized test procedure may not, in all circumstances, give a reliable indication of the actual risk of noise exposure during the typical use of a hand-held power sander. The aim of this work was to investigate the difference between the values declared by the manufacturers of hand-held power sanders and the measured noise values during actual use. The measurements of the equivalent sound pressure levels were carried out using an integrating–averaging sound level meter (B&K, model 2245) during the sanding of beech and spruce wood with different types of hand-held electric sanders (belt, random orbital, and orbital) with abrasives of coarse, medium, or fine grit. Upon comparing the measured and declared noise values, differences ranging from −6.3 dB to 19 dB(A) were identified for distinct sander types. The results of this study show that the use of declared noise emission values during risk assessments underestimates the magnitude of operator noise exposure.

**Keywords:** noise; emission; sander; wood; risk assessment

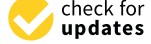


## 1. Introduction

Over the long term, noise-induced hearing impairment stands out as one of the most prevalent occupational diseases in Slovakia [1–3]. An employer utilizing or operating equipment that is a source of noise is obliged to ensure that technical, organizational, and other measures are taken to exclude or reduce, to the lowest possible and achievable level, the exposure of employees to noise as well as to ensure the protection of the health and safety of employees. To meet these obligations, the employer must evaluate the extent of noise exposure experienced by employees and, when deemed necessary, arrange for noise level measurements.

Noise emission information—the declared noise emission values given by machinery manufacturers—is one of the aspects that employers consider when assessing the risks from noise exposure. In accordance with their specific requirements and the provided information on equipment noise emission levels, the employer or user is tasked with selecting equipment that exhibits minimal noise emissions. This selection process should consider the prescribed limits and action values outlined in the applicable regulations [4]. This approach involves comparing the noise emission values measured for the evaluated machine with the values measured for similar machinery within the same group. Similar machinery refers to machinery designed to perform same function, with equivalent performance characteristics. The noise emissions of the compared machine devices must

be measured using a uniform testing procedure. The application of this approach must be based on an appropriate test procedure for noise measurement as well as on reliable and representative comparative noise emission data. The standard [5] offers details regarding the declaration of noise emission values. Additionally, it outlines the requirements for including information about product characteristics and acoustical data in the technical documentation for the declaration of noise emission. The standard also outlines the procedure for validating the declared noise emissions.

The hand-held power sander is a frequently used tool in woodworking, and it is a significant source of risk in terms of dust [6–12], vibration [13–17], and, not least, noise [18–21]. The noise reduction of a hand-held electric sander is an integral part of the design process and is achieved particularly by applying measures at the source to control noise. The success of the applied noise reduction measures is assessed on the basis of the actual noise emission values. The noise test code and requirements for the measurement of noise emissions of hand-held electric sanders are given in harmonized safety standards [22,23]. However, Patel and Brereton [24] investigated a restricted set of harmonized standards, utilized by machinery manufacturers to monitor the levels of noise emitted, and they identified that the noise hazards during the tools' intended uses were not accurately reflected in the noise emission values for sanders. Brereton and Patel [25] drew attention to the fact that the declared noise emission values obtained in accordance with a harmonized test procedure may not, in all circumstances, provide a reliable indication of the actual risk of noise exposure during typical use of the machinery. As an example, they gave noise emission values for hand-held electric wood sanders that were much lower than the values measured at the operator's position during actual use. Further, Patel and Hewitt [26] demonstrated that the actual usage levels for electric sanders exceeded the declared emission values by an average of $12 \pm 2$ dB(A), and for pneumatic sanders, the exceedance was $9 \pm 6$ dB(A). Similar conclusions were also reported by Shanks [27], who found an average difference of almost 9 dB when comparing between the declared values and the measured values during real operation.

It is important to distinguish between personal exposure to noise and noise emissions from hand-held electric sanders. The noise emissions from a sander, as determined under defined conditions, is an inherent characteristic of the machinery. Exposure of persons to noise from a sander depends on factors such as the technical condition of the power tool, the conditions of the use of the power tool, the characteristics of the workplace (e.g., noise absorption, noise scattering, or noise reflection), noise emissions from other sources (e.g., from other machinery), the position of persons relative to the noise sources, the duration of exposure to noise, and the use of hearing protectors [28]. Choosing practical operating conditions in accordance with the specifications outlined in the ISO 12001 standard [29], while ensuring consistent and reproducible test results, poses a challenge. The operating conditions for the determination of noise emission data for electrical sanders are specified in standard EN 62841-2-4 [23]. These tools are tested without an operator and under no-load conditions.

To the best of our knowledge, no previous studies have been undertaken to investigate the influences of sandpaper grit size and wood species on the noise levels of hand-held power sanders. The objective of this study was to investigate the difference between the values declared by the manufacturers of hand-held power sanders and the measured noise values during practical usage. We conducted measurements of the noise generated at the operator's ear for a sample of belt, random orbital, and orbital hand-held power sanders during a typical real-use task. Subsequently, these real-use noise levels were compared with the declared emission sound pressure levels provided for the same sanders to evaluate the reliability of the declared emission sound pressure levels in reporting noise hazards during the intended uses of hand-held power sanders.

## 2. Materials and Methods

### 2.1. Experimental Setup and Design

The experimental study was carried out in a test room with dimensions of 5 m (l) × 5 m (w) × 3.3 m (h), ensuring sufficient isolation from background noise. The test room had sound-reflecting walls (plastered bricks) and floor (concrete covered with linoleum), but the ceiling was sound-absorbing (cassettes made from perforated galvanized steel sheets). The time-averaged sound pressure level of the background noise as measured by a microphone was more than 15 dB below the corresponding uncorrected time-averaged sound pressure level when the tested sander was in operation. The design of the experimental setup is presented in Figure 1. A microphone was positioned at a distance of 0.2 m from the entrance of the operator's external ear canal and on the side of the most exposed ear. The ambient temperature and relative humidity were monitored using a microclimatic conditions monitor (Testo 480, Testo SE & Co., Titisee-Neustadt, Germany). Noise assessments were carried out at a temperature of 21 °C ± 1 °C and at a relative humidity of 38% ± 1%.

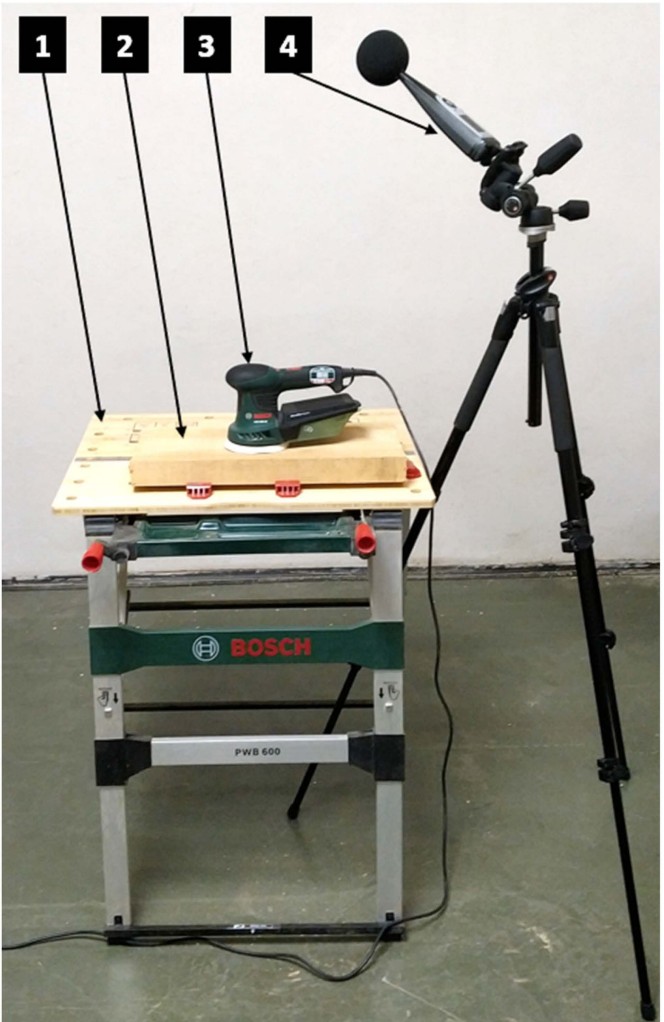

**Figure 1.** Experimental configuration design: 1—workbench, 2—test specimen, 3—sander, 4—sound level meter.

This research was conducted as a two-factor completely randomized experiment involving three different sandpaper grit sizes. Two types of wood species were investigated during the experiment. Five repetitions were performed for each treatment, resulting in a total of 150 runs of the test.

### 2.2. Description of Sanders under Test

The commercially available factory-fresh electric hand-held sanders under study (see Figure 2) were belt sanders (model PBS 75 A and professional model GBS 75 AE, Robert Bosch Power Tools GmbH, Stuttgart, Germany), random orbital sanders (model PEX 300 AE and professional model GEX 125-1 AE, Robert Bosch Power Tools GmbH, Stuttgart, Germany) and an orbital sander (model GSS 23 A, Robert Bosch Power Tools GmbH, Stuttgart, Germany). In case of belt sanders, abrasive belts (type LS309XH, Klingspor Schleifsysteme GmbH & Co. KG, Haiger, Germany) of three grit sizes (coarse, P60; medium, P120; and fine, P240) with dimensions 75 mm × 533 mm were used for sanding and were replaced after each trial. In the investigation of random orbital sanders, sanding papers with aluminum oxide abrasives with three distinct grit sizes (P60, P120, and P240) were utilized. A 125 mm diameter abrasive disc (PS 22 K, Klingspor, Bielsko-Biala, Poland) was substituted after each trial. The orbital sander was fitted with rectangular sanding sheets (type C 430, Bosch, Germany) of three different sanding grits (P60, P120, and P240) with dimensions of 93 mm × 186 mm. Sanding sheets were replaced after each trial.

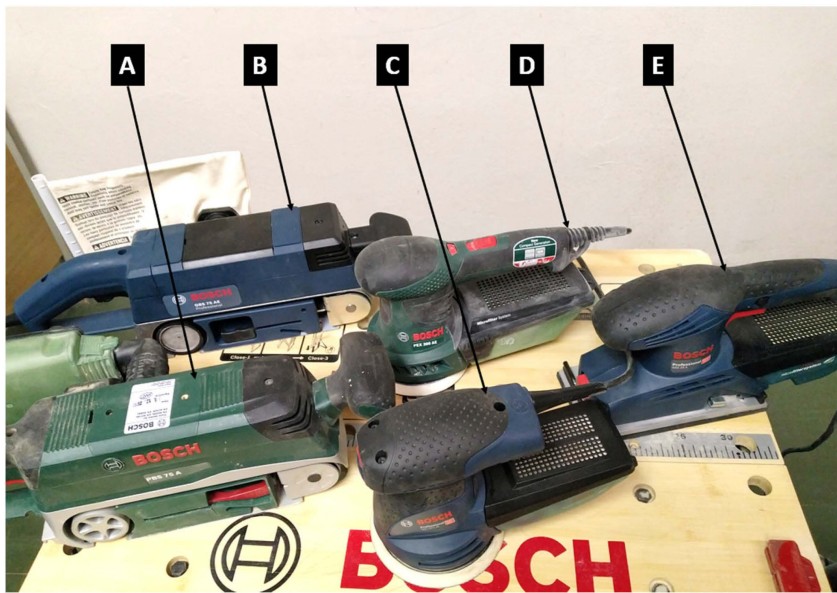

**Figure 2.** Electric hand-held sanders under study: A—PBS 75 A, B—GBS 75 AE, C—GEX 125-1 AE, D—PEX 300 AE, E—GSS 23 A.

The belt speed, stroke rate, as well as the sound power level and sound pressure level data, as specified in the technical information of sanders, are presented in Table 1.

**Table 1.** Noise and belt speed/stroke rate information provided in sander instructions.

| Sander | Belt Speed/Stroke Rate | Sound Power Level $L_{WA}$ (dB) | Sound Pressure Level $L_{PA}$ (dB) | Uncertainty K (-) |
|---|---|---|---|---|
| A | no-load belt speed: 350 m/min | 104 | 93 | 3 |
| B | no-load belt speed: 330 m/min | 96 | 85 | 3 |
| C | no-load orbital stroke rate: 24,000/min | 88 | 77 | 3 |
| D | | 96 | 85 | 3 |
| E | | 91 | 80 | 3 |

Moreover, the pressure force significantly influences the noise generated by a sander. This is substantiated by our methodologies for monitoring pressure force in our prior research endeavors [8,30]. In the current study, the sanding pressure's magnitude was

approximated based on the sander's weight and the surface area of the sandpaper. The estimated contact pressures for individual sanders are as follows: PBS 75 A—2791.1 Pa; GBS 75 AE—2960.0 Pa; GEX 125-1 AE—1038.9 Pa; PEX 300 AE—1197.9 Pa; and GSS 23 A—997.4 Pa.

### 2.3. Test Specimens

Investigated wood species used for the experiment were beech (*Fagus silvatica* L.) and spruce (*Picea abies* (L.) *H. Karst*). Test specimens in the form of planks of 500 mm × 250 mm × 50 mm dimensions were conditioned to a final moisture content of 10% before experimentation. According to the manufacturer's instructions, the moisture content of the test specimens was assessed using a wood moisture meter (model Testo 606-2, Testo SE & Co., Titisee-Neustadt, Germany) based on an electrical method (electrical resistance). A portable workbench (model PWB 600, Robert Bosch Power Tools GmbH, Stuttgart, Germany) was used for clamping the test specimens. Ex post analysis was performed to determine the density of the tested specimens according to standard ISO 13061-2:2014 [31]. Each specimen was measured with calipers with an accuracy of ±0.05 mm and weighted using a BP 3100 P balance (Sartorius AG, Goettingen, Germany) with an accuracy of ±0.01 g. These measurements were used to estimate the density of the wood. Table 2 summarizes more detailed information on wood species parameters.

**Table 2.** Investigated wood species parameters.

| Parameter | Specification |
| --- | --- |
| Species | beech (*Fagus silvatica* L.), spruce (*Picea abies* (L.) *H. Karst*) |
| Origin | Zvolen region, Central Slovakia, Supplier—University Forest Enterprise |
| Density | spruce ∼380 kg/m$^3$ <br> beech ∼590 kg/m$^3$ |
| Moisture content | The specimens were dried naturally at 20 ± 1 °C. |
| Fiber direction | 0° tangential cut/longitudinal tangential plane |

Note: all of the specimens were free from knots.

### 2.4. Noise Measurement Instrumentation

Noise measurements were performed with an integrating–averaging sound level meter (model 2245, Brüel & Kjær, Nærum, Denmark), which included a prepolarized condenser microphone (type 4966, Brüel & Kjær, Nærum, Denmark) and met the requirements of standard IEC 61672-1:2013 for Class 1 accuracy [32]. The meter's calibration was confirmed both before and after each set of measurements by a sound level meter acoustical calibrator (model 4231, Brüel & Kjær, Nærum, Denmark), as can be seen in Figure 3.

Either immediately before or after measuring the time-integrated sound pressure levels from the sander under test, the time-averaged sound pressure level of the background noise was obtained at the same microphone position and over the same integration time as those used for the measurement of the sander under test. The background noise measurements were made using a B&K hand-held sound analyzer Type 2245, fitted with a B&K microphone type 4966. Background noise levels in the test room were significantly lower than the noise generated by the sanders; the mean LAeq was 35 dB(A) when the sanders were not in operation.

### 2.5. Measurement Procedure

The measurement time interval of 3 min was derived from the time required to sand the test specimen under the following operating conditions: sander idle (15 s), sander at full load (165 s). A single individual, experienced in using the sander, executed the sanding task.

### 2.6. Data Analysis

Noise Work Partner software (version 1.6.3.0, Brüel & Kjær, Nærum, Denmark) was used for data post-processing. The normal distribution of the data was evaluated utilizing

the Shapiro–Wilk test. The *p*-values obtained from the Shapiro–Wilk test were calculated for each wood and grit type, revealing a significance level of $p \leq 0.001$. Consequently, it can be concluded that the data do not follow a normal distribution. Hence, a non-parametric Kruskal–Wallis one-way analysis of variance was conducted to investigate the relationship between grit size and the levels of noise generated during the sanding of beech and spruce wood. All statistical analyses were carried out using JASP computer software version 0.17.0 (University of Amsterdam, Amsterdam, The Netherlands).

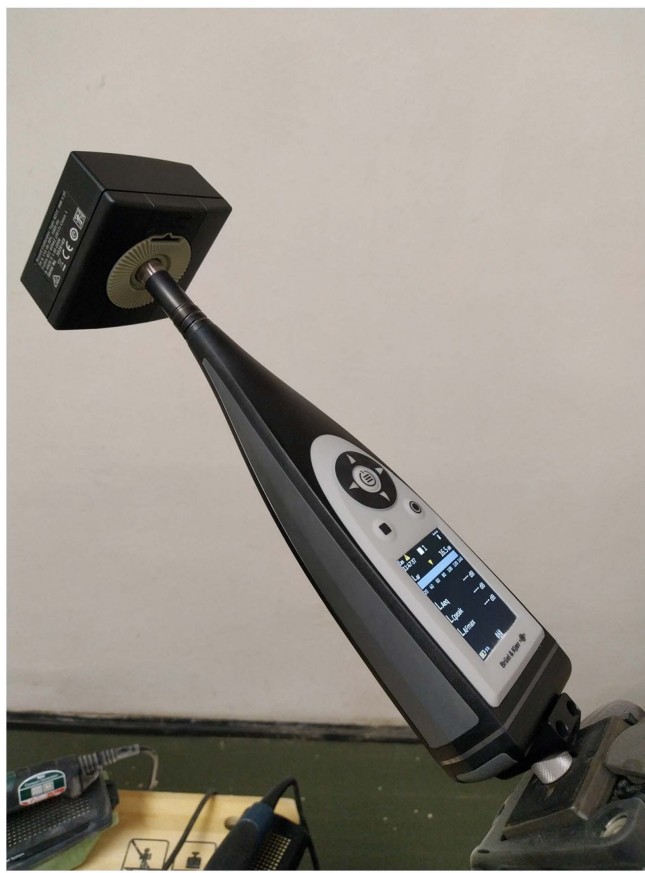

**Figure 3.** Calibration of integrating–averaging sound level meter.

### 3. Results

*3.1. Measured Values of Equivalent Noise Level at Sander Operator´s Ear*

The measured average values of the A-weighted equivalent continuous sound pressure level at the operator´s ear for the different types of hand-held electric sanders are shown in Table 3.

The Kruskal–Wallis test showed mean rank scores of 45.84 for P60, 33.52 for P120, and 34.64 for P240 of the measured equivalent sound pressure levels from beech wood sanding. Results of the test indicate that there is a non-significant ($p = 0.087$) difference in noise emissions generated during the sanding of beech wood by the examined grit types. Similarly, spruce wood data showed a non-significant ($p = 0.579$) difference in noise emissions. The mean rank scores for spruce wood were 41.66, 35.60, and 36.74 for sandpaper of P60, P120, and P240 grit, respectively. Our results demonstrated that the noise level during wood sanding did not depend on the grit of the sandpaper. Furthermore, we investigated the potential influence of wood type itself on noise emission values during the sanding process. By comparing the measured data from both beech wood and spruce wood sanding, no statistically significant difference ($p = 0.924$) between these two wood species was observed. Executed Kruskal–Wallis tests results are summarized in the following Table 4.

**Table 3.** Measured values of the A-weighted equivalent continuous sound pressure level ($L_{Aeq}$) at sander operator´s ear (arithmetic mean ± standard deviation, n = 5).

| Sander | Sandpaper Grit | Wood Species | Sound Pressure Level $L_{Aeq}$ (dB) |
|--------|---------------|--------------|-------------------------------------|
| A | P60 | beech | 90.3 ± 0.4 |
|  |  | spruce | 89.9 ± 0.3 |
|  | P120 | beech | 89.3 ± 0.2 |
|  |  | spruce | 89.9 ± 0.3 |
|  | P240 | beech | 89.3 ± 0.1 |
|  |  | spruce | 89.8 ± 0.3 |
| B | P60 | beech | 90.7 ± 1.1 |
|  |  | spruce | 90.0 ± 0.6 |
|  | P120 | beech | 90.0 ± 0.9 |
|  |  | spruce | 89.0 ± 0.3 |
|  | P240 | beech | 89.7 ± 0.4 |
|  |  | spruce | 89.9 ± 0.9 |
| C | P60 | beech | 82.4 ± 0.5 |
|  |  | spruce | 82.3 ± 0.7 |
|  | P120 | beech | 81.0 ± 0.6 |
|  |  | spruce | 81.5 ± 0.6 |
|  | P240 | beech | 81.3 ± 0.5 |
|  |  | spruce | 81.4 ± 0.3 |
| D | P60 | beech | 90.5 ± 0.9 |
|  |  | spruce | 89.4 ± 0.5 |
|  | P120 | beech | 89.3 ± 0.4 |
|  |  | spruce | 88.9 ± 0.8 |
|  | P240 | beech | 89.8 ± 0.7 |
|  |  | spruce | 88.6 ± 0.3 |
| E | P60 | beech | 81.3 ± 0.5 |
|  |  | spruce | 83.2 ± 0.9 |
|  | P120 | beech | 81.2 ± 0.6 |
|  |  | spruce | 83.5 ± 0.4 |
|  | P240 | beech | 81.1 ± 0.3 |
|  |  | spruce | 83.8 ± 0.9 |

**Table 4.** Kruskal–Wallis test results examining the impact of grit size and wood type on equivalent sound pressure levels.

| Wood Type | Factor | Statistic H | df | *p* Value |
|-----------|--------|-------------|-----|-----------|
| beech | grit size | 4.891 | 2 | 0.087 |
| spruce | grit size | 1.093 | 2 | 0.579 |
| - | wood type | 0.009 | 1 | 0.924 |

The raincloud plots shown in Figure 4 provide a comprehensive overview of the noise emission data for the different investigated scenarios of sandpaper grit and wood type used for sanding. Each raincloud plot consists of three components: a density plot, a box plot, and individual data points. From the raincloud plots, it is evident that there is no significant variation in equivalent sound pressure levels across different sandpaper grits or wood types. The density plots show overlapping distributions, and the box plots demonstrate similar mean values for all conditions. This confirms that neither sandpaper grit nor wood type has a pronounced effect on the noise generated during wood sanding.

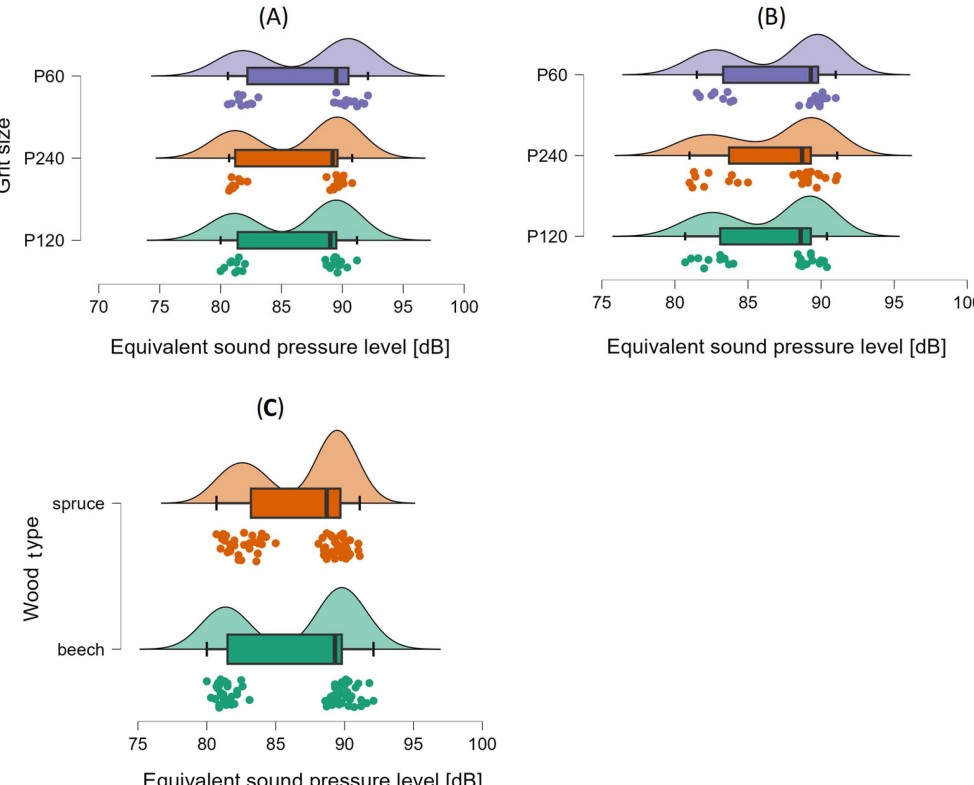

**Figure 4.** Raincloud plots illustrating the impact of sandpaper grit and wood type on the equivalent sound pressure level during wood sanding: (**A**) beech wood sanding, (**B**) spruce wood sanding, (**C**) wood type comparison regardless of grit size.

*3.2. Comparing Simulated Real-Use and Declared Noise Data*

The difference between the measured real-use values and the declared emission values, taking uncertainty into account, is plotted in Figure 5. A value exceeding zero suggests that the declared value may result in under-protection when compared to the real-use values. The range was from −6.3 to 1.9 dB(A). In three out five cases, the declared emission underestimates thein real-use data.

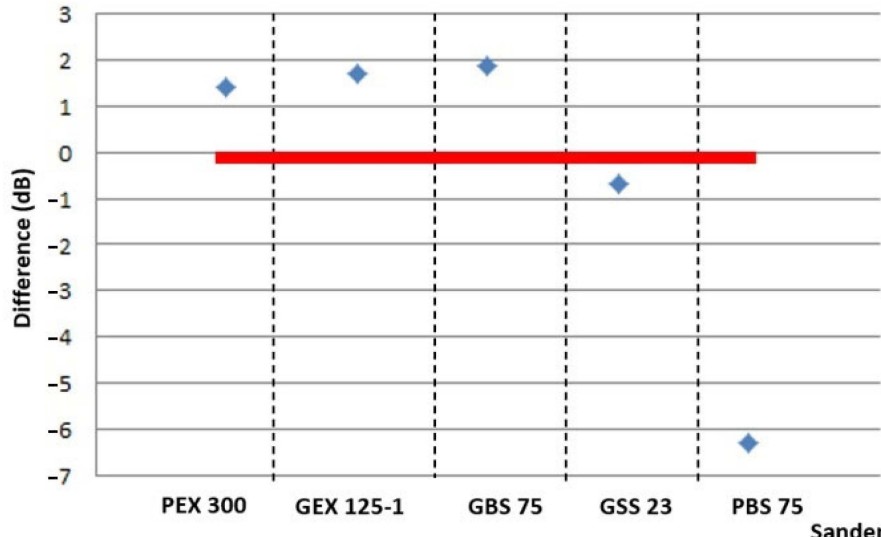

**Figure 5.** Difference between the measured noise data and the manufacturer's declared noise values for each sander.

## 4. Discussion

Manufacturers, importers, and suppliers of hand-held sanders must adhere to the noise regulations outlined in the Machinery Directive 2006/42/EC [33]. The Machinery Directive requires the provision of A-weighted emission sound pressure levels at workstations ($L$pA); a A-weighted sound power level ($L$WA) may be required based on the emission $L$pA value. Hence, the anticipated procedure involves initially measuring the emission $L$pA, which is expected to signify hazards and facilitate a comparison among competing machines. Nevertheless, for hand-held power tools like sanders, manufacturers are mandated by the harmonized noise test code to derive $L$WA, from which emission $L$pA values are computed. According to Patel and Brereton [24] emission $L$pA values may not represent real-use levels for the following reasons: the sound pressure level on the surface at a distance of 1 m may not accurately reflect the noise experienced at the operator's position, and the operating conditions may not precisely capture the loudest operation in the typical use of the machine. The standard [22] acknowledges that the emission sound pressure level, as determined by the method outlined in said standard, typically tends to be lower than directly measured sound pressure levels for the same machine in a typical workroom environment. This discrepancy is attributed to the impact of sound-reflecting surfaces in the workroom, in contrast to the open-field conditions specified in the test. Observable variations generally range from 1 to 5 dB, although in exceptional instances, the difference may be more significant [22]. The noise test code for electrical sanders calculates the sound power level based on measurements of the surface sound pressure level. In the original version of the noise test code, the emission $L$pA was equivalent to the surface sound pressure level at a distance of 1 m from the tool. In typical usage, sanders are often positioned significantly closer than 1 m to the operator's ear. For this reason, in April 2022, amendment A11 was introduced to modify standard EN 62841-1:2015/A11:2022 [22] and to correct the determination of the emission sound pressure level for hand-held tools. The aforementioned distance was changed to 0.7 m and the experimentally determined quantity Q was changed from 11 to 8.

According to standard ISO 12001, the noise test code shall specify an operating condition that is reproducible and is representative of the noisiest operation in typical usage of the machine under test. The operating conditions for the determination of noise emission data for electric hand-held sanders are specified in standard [23]. These tools are suspended with the plate of the tool kept horizontal and tested under no-load conditions. Our results confirmed the correctness of these specifications. We observed that in several cases (e.g., see Figure 6), the no-load part of tests represented a noisier operating condition than an actual sanding task, especially in the case of belt sanders. This phenomenon has already been identified in other areas, e.g., in relation to circular saw noise [34] or chainsaw vibrations [35]. A potential explanation for this phenomenon might be that aerodynamic noise at idling, resulting from turbulent flow generated by the movement of the sanding belt in the air, is more pronounced, whereas during sanding, it is mitigated by the presence of the workpiece.

The actual usage levels, as indicated in Table 3, highlight that sanders can be relatively noisy, with noise levels reaching up to 91 dB(A) at the operator's ear. Comparing the noise levels of different sanders, based on the declared values as well as our results, we can conclude that belt sanders emit the most noise. As outlined in the safety standard for electric sanders [22], the user manual is required to indicate that the declared noise emission values are suitable for an initial assessment of exposure. Nevertheless, the data presented in Figure 5 suggest that this guidance may not be applicable to most sanders. Preliminary noise exposure assessments relying on these declared emission $L$pA values are prone to underestimating the actual noise risk during real usage, potentially misleading users about the necessary precautions to control the risk effectively.

Several limitations of this study need to be considered. Firstly, the relatively small sample of sanders imposes limitations on the generalizability of the obtained results. It is unfortunate that the study did not include sanders from several manufacturers. Secondly,

it is important to acknowledge a minor limitation in this study, which arises from the inability to verify the declared values of the tested sanders due to the absence of technical equipment. Thirdly, a potential source of weakness in this study that could have influenced the measurement of noise was the lack of precise control over the magnitude of the feed force during the sanding task.

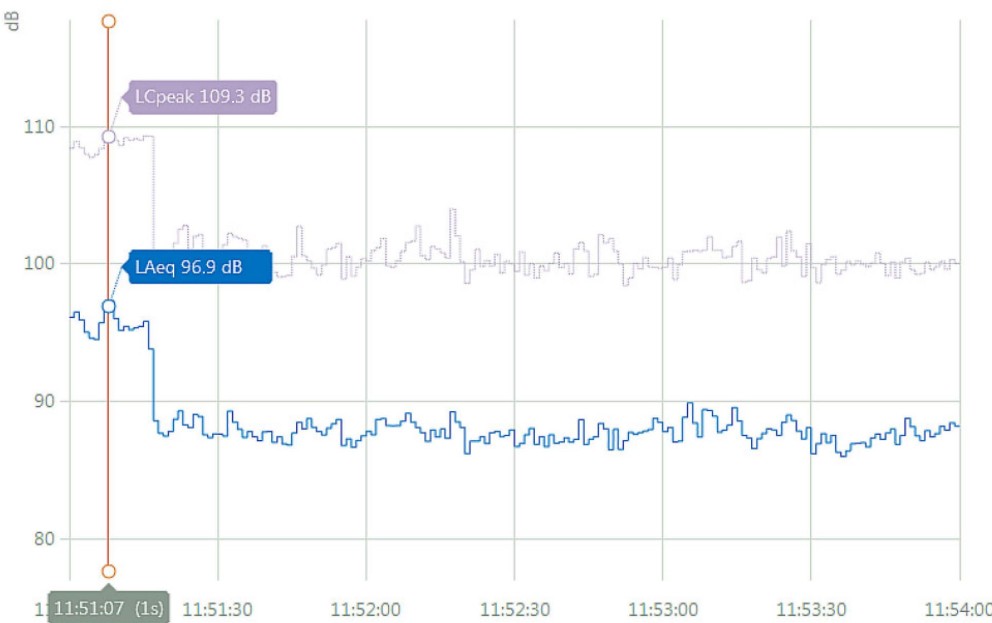

**Figure 6.** Time domain of sound pressure level values during sanding spruce wood with sander PEX 300 AE.

## 5. Conclusions

Manufacturers of machinery must disclose airborne noise emissions to enable verification of cutting-edge noise control by manufacturers, market surveillance authorities, and other entities. This declaration serves multiple purposes, including assisting users in identifying tools and machines with low or reduced noise levels, representing noise hazards during the intended uses of these tools and machines, and informing employers' noise risk assessments [24].

This paper investigated the differences between the values declared by manufacturers of hand-held power sanders and the measured noise values in actual use. Based on experimental results, the following findings are concluded for this specific case:

- Declared noise emission values in risk assessments underestimate the magnitude of operator noise exposure;
- No statistically significant differences were found when the coarse-grit sandpaper was compared to medium-grit and fine-grit varieties in terms of emitted noise during sanding tasks;
- By comparing the measured noise data from both beech wood and spruce wood sanding, no statistically significant difference between these two wood species was observed.

**Author Contributions:** Conceptualization, M.D.; methodology, M.D. and M.S.; investigation, M.D. and R.J.; statistical analysis, J.S.; writing—original draft preparation, M.D. and J.S.; writing—review and editing, J.S. and R.H.; visualization, M.D.; supervision, M.S. All authors have read and agreed to the published version of the manuscript.

**Funding:** This research was funded by research contract KEGA grant number 009TU Z-4/2022.

**Data Availability Statement:** The data presented in this study can be found within the article.

**Acknowledgments:** This paper is based on work performed under research contract KEGA 009TU Z-4/2022 of the Cultural and Educational Grant Agency of the Ministry of Education, Science, Research and Sports of the Slovak Republic and the Slovak Academy of Sciences, whose support is gratefully acknowledged.

**Conflicts of Interest:** The authors declare no conflict of interest. The funders had no role in the design of the study; in the collection, analyses, or interpretation of data; in the writing of the manuscript; or in the decision to publish the results.

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
