# Peer review of "Comparison of the Declared and Simulated Real-Use Noise Data during Wood Sanding Using a Hand-Held Power Sander"

_acoustics, doi:10.3390/acoustics5040064_

Round 1

Reviewer 1 Report

Comments and Suggestions for Authors

Dear Author, I have read your manuscript very carefully. This work has a correct structure and a clearly defined aim of study. However I see in them huge gaps in intruduction and methodology of research. 

The introduction is very weak and does not address many aspects of noise analysis of the cavity treatment process. Noise analysis is also used to monitor process quality and tool wear, and this would also be worth mentioning in the introduction. This should especially have an emphasis on wood-related processes. 

A great gap in the description of the survey methodology is the lack of basic information on the samples of wood analysed. Only the species is given. Wood is a very complex and advanced material and many parameters of its structural structure (density, width of annual increments, proportion of late wood in annual increments, knots, fibre direction, machining plane regarding to wood core etc.) or physical conditions (wood moisture content, wood temperature, drying method, etc.) and its origin affect the implementation of machining processes and, consequently, noise. 

A major deficiency in the test methodology is also the lack of information on the contact force of the hand grinders against the material and the speed of their feed movements, if any. These parameters also have a significant impact on the noise generated during the execution of the wood machining process and not only.

There is also a lack of information about the baseline noise measurement in the room without switching on the equipment. And how noise was analysed including information on idling of equipment. Only the information that idling was measured is given. 

In addition, there is no information on the hand-held grinders used for the analysis, namely at what frequency did they perform vibration movements and were these values similar?

Reviewer 2 Report

Comments and Suggestions for Authors

This manuscript discusses a study conducted on hand-held power sanders, which are commonly used in woodworking. The study aimed to compare the declared noise emission values provided by the manufacturers of these tools with the actual noise values measured during the use of sanders. The authors noted that differences between the declared values and the actual values ranged from -6.3 dB to 19 dB(A) depending on the type of sander. This means that relying on declared noise values does not accurately reflect the real risk of noise exposure during typical use of these tools. The study was intended to raise awareness among users that the actual noise level may be higher than what manufacturers suggest.

The material takes the form of a report describing the step-by-step research methodology used to compare the noise emitted by the devices. The research was conducted meticulously, varying the sandpaper grades, and using two types of wood.

My suggestions for the authors:

·         In the study, no significant differences in emitted noise were observed for different working configurations (sandpaper type, wood type). In my opinion, based on the presented results, a key factor for assessing emitted noise could be monitoring the load on the wood sander, understood as reading current parameters (e.g., total power drawn from the electrical energy source by the set of connected energy consumers).

·         When conducting this type of research, it's important to ensure proper soundproofing (using sound-absorbing foam). Conducting research in rooms with walls can lead to reflective phenomena, thus complicating the research.

·         Where possible, a consistent style of presenting results (charts) should be adopted, such as using data export capabilities to generate charts. This can improve the clarity and aesthetics of the work.

·         Analyzing the spectrum of emitted noise can help identify frequencies related to the emitted noise, allowing for a better interpretation of the noise generation mechanism. Consequently, it may be possible to identify the cause of discrepancies between the results provided by manufacturers and the measured values.

In the end, the paper addresses a very interesting phenomenon related to safety and occupational hygiene. The material is presented in a clear and technically correct manner.

Reviewer 3 Report

Comments and Suggestions for Authors

This study will measure hand sander equipment noise in a variety of sander types with different sized sand-paper to compare noise measured to the manufacturer's noise emission declarations.

Introduction:

line 43 - authors may want to remove the 2nd "machinery" word.

Materials and Methods:

When manufacturers measure noise, is it under the exact same conditions (i.e., same absorption in walls, floors and ceilings). Do they measure also near the operator's ear? Were the author's measurements and the manufacturer's measurements performed at the same distance, i.e., .7meters from the sander?

Discussion: line 274 - remove the second "the"

Round 2

Reviewer 1 Report

Comments and Suggestions for Authors

Dear Authors, 

thank you for your responses to my review. Most of them are acceptable, however, there are issues that I cannot accept. One of these is the material analysed. The authors agreed with the reviewer that wood is a complex material. Each piece of wood is different, even those from the same tree. This is why it is so important to give precise data about the analysed wood, above all the species, origin, density, average number of annual growths, moisture content (preferably determined by the drying and weighing method), fibre direction. This information is missing in the reviewed paper and this is unacceptable. 

The second issue is the different, unquantified pressure force of the grinders, this is also a variable that can have a huge impact on noise and which was not included in the methodology. These issues need to be corrected.

Round 3

Reviewer 1 Report

Comments and Suggestions for Authors

Dear Authors,

thanks again for your cooperation during review of this paper. In my opinion reviewed paper in current form can be published in Acoustics journal